# Attitudes and Beliefs of Mental Health Professionals towards Trans People: A Systematic Review of the Literature

**DOI:** 10.3390/ijerph20156495

**Published:** 2023-08-01

**Authors:** María Asunción Cutillas-Fernández, José Antonio Jiménez-Barbero, María Herrera-Giménez, Luis Alberto Forcén-Muñoz, Ismael Jiménez-Ruiz

**Affiliations:** 1Psychiatrist Servicio Murciano de Salud, 30100 Murcia, Spain; 2Department of Nursing, Faculty of Nursing, University of Murcia, 30120 Murcia, Spain; 3ENFERAVANZA, Murcia Institute for BioHealth Research (IMIB-Arrixaca), 30120 Murcia, Spain

**Keywords:** transgender, professionals, mental health, attitudes, beliefs

## Abstract

A systematic review was conducted to assess and synthesize recent research on mental health professionals’ attitudes towards trans people. The main objectives of our research were (a) to identify, synthesize, and analyze the scientific evidence available so far about the attitudes of mental health professionals towards the trans community, and (b) to determine the factors related to these professionals’ attitudes, paying special attention to psychosocial and cultural aspects. A systematic search was carried out in the following electronic databases: Pubmed, Web of Science, PsycINFO, PsycARTICLES, Gender Studies Database, and Lilacs. A total of 32 articles of quantitative (n = 19), qualitative (n = 11), and mixed (n = 2) design, published up to March 2023, were included. Most studies used a cross-sectional or qualitative design, limiting the possibility of generalizing the results. The studies reviewed indicated mostly positive attitudes among the professionals, depending on their psychosocial characteristics. In line with the results of our review, we recommend that the training of professionals is important to improve their positive attitudes towards transgender and gender diverse people.

## 1. Introduction

Several terms have been used during the last decades in reference to individuals whose gender identity is not related to the sex assigned at birth, including transgender, transsexual, trans, gender fluid, and gender diverse, among others [1,2]. These people may identify as male or female or identify outside the gender binary [3]. 

When these people grow up in cis-normative environments and societies and do not meet the expectations generated about their assigned sex, they are subject to discrimination and inequality throughout their lives. Among other inequalities, we can highlight the barriers to socio-health care [4,5]. 

These barriers, which range from individual obstacles by the professionals themselves to organizational barriers, cause difficulties both in access to the health care systems and the care itself in health and disease processes [6,7,8,9]. In this respect, a meta-analysis by Zeeman et al. (2018) reports that LGBTI people are more likely to experience health problems, inequalities due to heteronormativity or heterosexism, minority stress, and experiences of victimization and discrimination, aggravated by stigma [10]. 

Although some guidelines have been proposed to ensure minimum multicultural skills among mental health professionals [2], they have not been incorporated into their training programs and many professionals are unaware of their existence [11,12]. As a consequence, transgender people continue to find mental health services unpleasant, and many are wary of accessing these services, due to their fear of being revictimized or hurt by professionals [1,3,13].

Various authors have defined attitudes or beliefs as a person’s favorable or unfavorable evaluation of an object or social situation in terms of the consequences of their behavior toward it [4,10]. The difference between attitude and belief is that, although both concepts share a cognitive dimension, attitudes are essentially affective phenomena [14,15]. Following Escámez’s definition, attitudes are determined by three main components: cognitive, which denotes beliefs and knowledge; affective, which refers to feelings or the evaluation made by the person; and, finally, behavioral, in relation to the behavior in the presence of the object [14,16]. 

To this day, studies focusing on the attitudes and beliefs of professionals are still limited. Most span the entire LGBT spectrum [17,18,19,20,21,22,23,24] and tend to focus exclusively on the medical community [6,10,17]. Furthermore, we must highlight the great heterogeneity found regarding the conceptualization of the term “attitude” among the different instruments used to measure attitudes among health professionals, which makes it difficult to interpret and generalize the results of these instruments. The complex interrelationships of trans individuals with the health system make specific research in mental health necessary to specifically improve the overall health of this collective. 

Although systematic reviews have been published that analyze the attitudes and beliefs of health professionals towards transgender and gender diverse people [25], this is the first one to specifically evaluate the attitudes and beliefs of mental health professionals, defined as psychiatric physicians, mental health nurses, psychologists, social workers, and occupational therapists, as well as the factors that influence these attitudes. 

### 1.1. Objectives

The main objective of this study was to identify, synthesize, and analyze the scientific evidence available so far about the attitudes and beliefs of mental health professionals towards the trans community. As a secondary objective, we proposed to study which variables are related to the attitudes of these professionals, paying special attention to their psychosocial, training, and cultural aspects.

To address this goal, the following PICO (or Patient/population), Intervention, Comparison, Outcomes) question was asked: “What attitudes/beliefs do mental health service professionals have towards trans people?”

### 1.2. Methods

The protocol for the elaboration of this systematic review was carried out following the recommendations of PRISMA 2020 [26]. From its inception, this study was recorded in PROSPERO (International Prospective Register of Ongoing Systematic Reviews), where it appears with registration number 177761. URL (accessed on 10 July 2023): http://www.crd.york.ac.uk/prospero.

The methods used were pre-specified and documented in a protocol available online. URL (accessed on 10 July 2023): https://www.crd.york.ac.uk/prospero/display_record.php?ID=CRD42022177761.

### 1.3. Search Strategy

A systematic search was carried out in the following electronic databases: Pubmed, Web of Science, PsycINFO, PsycARTICLES, Gender Studies Database, and LILACS. The descriptors used were: “trans*” OR “transgendered person” OR “transgender person” OR “gender dysphoria” OR “gender non-conforming” AND “attitudes” OR “beliefs” OR “practices” OR “stereotyped” OR “knowledge” AND “psychologist” OR “psychotherapist” OR “psychiatrist” OR “nursing” OR “mental health services” OR “mental health professionals”. The complete strategy used can be found in Appendix A. No temporal or language restrictions were established. The last access to the information sources was made in the Web of Science database on 3 March 2023. 

The search was conducted by two independent researchers who made lists of potentially eligible articles. This list was subsequently revised, and disagreements were resolved through the intervention of a third reviewer. 

### 1.4. Inclusion and Exclusion Criteria

We used the following criteria for the inclusion of studies in this review: (a) the study group included professionals who were active in centers, institutions, or services specialized in mental health (psychiatric physicians, mental health nurses, psychologists, social workers, and occupational therapists); (b) at least one of the main objectives of the study was to determine the attitudes of such professionals towards trans people; (c) the studies used a quantitative (observational, experimental, or quasi-experimental), qualitative, or mixed design.

The exclusion criteria were: (a) studies whose main objective did not focus on the attitudes of mental health professionals; (b) studies exploring patients’ perceptions of professionals’ attitudes; (c) secondary studies (narrative or systematic reviews). 

### 1.5. Selection of the Studies

Study selection was carried out in two phases, following the indications of the PRISMA statement [26].

The first and second authors independently screened potentially eligible articles using a checklist that met the criteria established by the protocol, according to the PICO question. The preselected studies were agreed upon and discrepancies were resolved by a third reviewer. Accepted articles were carefully read and their bibliographic reference lists were examined to identify possibly relevant articles not located in the initial search (see Figure 1).

### 1.6. Analysis of Risk Bias

The selected studies were submitted to risk-of-bias analysis, which was performed by two independent reviewers. The instruments used for this analysis were the statement of the STROBE initiative for observational studies [27], as well as the template proposed by the Critical Appraisal Skills Programme for the critical reading of qualitative studies [28]. The scale contains ten items that assess the research objectives, qualitative methodology and research method, and the reflexivity and ethical aspects of the study. The cut-off point for the eligibility of the studies was established at the mean value of each scale, that is, the article had to exceed 50% of items on the evaluation scale to be included in the systematic review. The scoring was carried out as follows: 1 point if the criterion was fully met, half a point if the item was partially met, and 0 if it did not appear. In cases where no consensus was reached about the acceptability of an article, a third reviewer was consulted. Finally, interjudge reliability was calculated using intraclass correlation analysis. 

### 1.7. Tabulation and Data Analysis

The articles selected for our review were coded by the main author in an Excel database. This coding was reviewed by the second and third authors of the study. Discrepancies were resolved by consensus among all authors. Subsequently, summary tables were created in which the relevant data from each study were recorded according to the following categories:-For quantitative studies: Study reference, country in which it was carried out, design, objective, sample characteristics, outcome measures and measurement instruments, main findings, and conclusions.-For qualitative studies: Study reference, country in which it was carried out, objectives, sample characteristics, sources of information, method of analysis, categories, and conclusions.

### 1.8. Data Synthesis

Due to the high heterogeneity found in the outcome measures provided by the quantitative studies, we ruled out performing a meta-analysis on them. Instead, we conducted a narrative synthesis of the results, in which we summarized the characteristics of the study populations, measures, and interventions, using descriptive statistics [29].

In the case of the qualitative studies, the categories obtained in the different studies were grouped into common themes from which the narrative synthesis of the results was developed. The grouping of the categories was agreed upon by the third and fourth authors after exhaustive reading of the articles and agreed with the rest of the research team. 

## 2. Results

As shown in Figure 1, the initial search found 24,492 potential publications. After reading titles and abstracts, 24,277 papers were excluded and another 3 articles located by reverse search were included. In the second phase of selection, after reading the full texts, 90 articles were excluded for not meeting the inclusion criteria established in the checklists. Subsequently, the risk-of-bias analysis excluded 7 articles for failing to meet the established methodological quality criteria. The scores given by each reviewer to each of the accepted studies, as well as the final score obtained by consensus, are available online, in a document annexed to the protocol. The intraclass correlation coefficient showed high interjudge reliability for peer analysis of the risk of bias (CCI = 0.87, *p* = 0.000).

The systematic review finally included 32 articles, of which 19 presented a quantitative design, 11 a qualitative design, and 2 a mixed design. Methodological differences, as well as the diversity of the objectives that these designs presented, recommended analyzing them separately [30].

### 2.1. Quantitative Studies

The 19 quantitative studies were published between 2010 and 2022, except for one article published in 1986 The study sample had the following characteristics: the age range was between 18 and 93 years, as one study included university students. The mean age could not be determined because several authors omitted this information. Most of the people included in the selected studies identified with the female gender, although one of the articles did not provide this datum (Table 1). The minimum and maximum sample sizes were 53 and 2850, respectively. The most commonly used study design was cross-sectional design (n = 13), followed by experimental or quasi-experimental studies (n = 4), and control cases (n = 2). The duration of the interventions evaluated by the studies included in the systematic review varied between 3 months and 4 years [8,12,17,31,32,33,34,35,36,37,38,39,40,41,42,43,44,45,46].

Given the high heterogeneity of the outcome measures used in the included studies, we decided to group them into categories for analysis. The following categories were established that correspond to the main and secondary objectives set out in our study. For the quantitative studies, the following categories were established: (1) attitudes, behaviors, and beliefs about trans people; this category answered the research question that guided the main objective of this study; (2) sociodemographic variables; (3) professionals’ training, knowledge, and skills; (4) stigma, social distancing, and transphobia.

#### 2.1.1. Attitudes and Beliefs about Trans People

We found 12 studies that addressed the attitudes of professionals towards trans people. Overall, the studies reviewed found a trend towards positive (i.e., supported the basic human rights of trans people) and affirmative attitudes among professionals [17,31,32,33,34,35,36,37,38]. Three of the studies [17,36,39] used the GTS scale [47] to evaluate the cognitive, affective, and behavioral components of negative attitudes. One of these studies [36] found that mental health professionals had relatively more positive attitudes than the general population. Along this line, Franzini and Casinelli (1986) had concluded in earlier years that clinical psychologists have a more favorable position than the rest of the groups surveyed [32].

On the other hand, Johnson and Federman (2014) found more positive attitudes among young therapists, suggesting a likely greater impact of early training on sexual diversity and the long-term impact on their professional career [33]. More critical is the study of Francia-Martínez et al. warning that up to 16.4% of the surveyed professionals said they did not feel comfortable working with trans people [40]. Two of the studies observed higher values for extreme prejudice among those professionals who attributed gender diversity to a psychological, ethical-moral, or religious cause [31,41].

#### 2.1.2. Sociodemographic Characteristics Associated with Positive and Negative Attitudes

Six cross-sectional studies explored the influence of sociodemographic characteristics on professionals’ attitudes towards trans people [8,33,37,41,42,43].

The sociodemographic characteristics (a) belonging to a racial or sexual minority [8,37,41] and (b) having a relationship with trans people outside the professional field [37,41,42] were correlated with positive attitudes.

Regarding demographic differences, Johnson and Federman (2014) related being younger and residing in politically progressive states (reliably voted Democratic in presidential elections in the USA) with a greater interest in receiving specific training [33]. The study of Gaspodini and Falcke (2018), however, found no significant differences in the attitudes of professionals residing in rural areas compared to those residing in cities [41]. Concerning age, most studies found no significant results [8,42,43]. Only one study [33] found that psychologists born after 1969 had affirmative attitudes and more specific training.

In contrast, religiosity [37,42], residing in a politically conservative state (reliably voted Republican in presidential elections in the USA), and being older [33] were identified as variables that could interfere negatively.

#### 2.1.3. Professionals’ Training, Knowledge, and Skills

Three studies, two pre-experimental and one cross-sectional [33,38,44], assessed the influence of training on professionals; the study of Craig et al. (2015) noted that the professionals positively valued the training provided and were more open to changing their behaviors after receiving it [44].

Regarding quasi-experimental studies focused on evaluating the impact of training on the professionals’ attitudes, they highlighted the potential of training to reduce professionals’ negative attitudes and their impact on health care for trans people, pointing to training as the first step to improve the health of this group [31,37]. Lelutiu-Weinberger and Pachankis (2017) also pointed out the potential of training to teach professionals how to become agents of change and of reduction of stigma [45].

Three cross-sectional studies addressed professionals’ knowledge about consensual data on trans people [32,36,40]. In general, this knowledge was related to positive attitudes. The study of Franzini and Casinelli (1986) found no differences in knowledge between mental health professionals and other medical specialties [32]. Similarly, in a more recent study, Francia-Martínez et al. (2017) concluded that a high number of professionals were unaware of important aspects such as the difference between sexual orientation and gender identity or the different needs of a trans user compared to a homosexual or bisexual user [40]. In contrast, Willoughby et al. (2010) found acceptable knowledge among the professionals evaluated (i.e., the test on knowledge of transgenderism yielded a positively skewed distribution, with scores falling in the higher range, indicating higher knowledge of transgenderism) [36].

Three cross-sectional and one mixed study addressed professional competencies as an outcome measure [8,12,33,43]. The study of Whitman and Han (2017) noted that those professionals who strive to know their own limitations and biases were less likely to perpetuate stigma towards trans people, both explicitly and implicitly [12]. The study of Dispenza and O’Hara (2016) reached similar conclusions, suggesting greater competence for working with trans and gender diversity people in professionals who belong to a sexual minority [42]. Johnson and Federman (2014) stressed that the psychologists evaluated defined themselves as competent to treat LGBT people, even though they recognized the scarce formal education received [33]. Similarly, the study of Riggs and Bartholomaeus (2016) noted that the men surveyed had more experience working with trans people but less formal training [8].

#### 2.1.4. Stigma, Social Distancing, and Transphobia

Two studies, one cross-sectional and one of mixed design [12,43], included stigma among the professionals in their studies. Vijay et al. (2018) found a positive association between concepts related to stigma (prejudice towards trans people, internalized shame, fear of trans people, beliefs about health care for the collective, and stereotypes) with a tendency towards discrimination by professionals [43]. Whitman and Han (2017) decried that, despite the fact that the participants surveyed reported being mostly aware of the ethical issues connected with stigmatizing trans identities as mental disorders, almost a quarter of the participants had pathologizing beliefs about them (e.g., that gender diverse identities represent mental disorders or spiritual flaws) [12].

Five studies addressed social distancing and transphobia towards the trans collective [31,39,40,43,46]. Their conclusions indicated that the training received by mental health professionals in which the health needs of the trans group were addressed could be useful to reduce these behaviors. On another hand, Francia-Martínez et al. (2017) found a moderate or low level of social distancing among the professionals surveyed, although around 40% of them said that, although they would maintain a friendly relationship with a trans person, they would not live under the same roof or in the same room with them [40].

### 2.2. Qualitative Studies

We included 11 qualitative studies published between 2012 and 2019 [48,49,50,51,52,53,54,55,56,57,58]. The age range of the participants was between 24 and 76 years. The mean age of the sample could not be determined because many of the studies omitted this datum. Similarly, the sex ratio of the sample could not be accurately determined because some articles did not provide this information. In those cases where it was reported, it was included in the results table (Table 2). The minimum and maximum sample sizes were 8 and 110, respectively. The methods of analysis used were thematic analysis (n = 2), grounded theory (n = 4), Weiss approximation method (n = 1), content analysis (n = 3), and phenomenological analysis (n = 1).

Given the high heterogeneity of the categories established by the studies included in this review, we decided to group them into conceptual themes, as recommended by Percy et al. (2015) [59]. The following themes were therefore established: (a) attitudes and beliefs towards trans people; (b) socio-demographic characteristics of professionals; (c) training, experience, and competences of professionals; (d) accessibility of trans people to the health system; and (e) therapeutic tools for working with trans people.

#### 2.2.1. Attitudes and Beliefs towards Trans People

Five of the selected studies addressed the attitudes and beliefs of mental health professionals about trans people [49,51,54,55,56]. In general, professionals’ positive attitudes were related to causal attributions about the psychosocial nature of gender identity. Along this line, the study of Gaspodini and Falcke (2018) identified positive attitudes among professionals who attributed gender differences to psychosocial variables; that is, among those who attributed a multifactorial cause rather than a biological, psychological, or ethical-moral cause [55]. Similarly, the study of Whitehead et al. (2012) focused on the qualities presented by the professionals who defined themselves as “trans-friendly”, finding important differences regarding gender ideology or causal attributions of gender differences [54].

Both the study of Lefkowitz Ayla (2017) and that of Whitehead et al. (2012) highlighted that most of the professionals interviewed had a binary representation of gender, considering trans people as individuals who are medically transitioning to another gender [51,54]. Likewise, health was attributed to the correspondence of gender and anatomical sex. Conversely, they believed that being a transgender person had a negative impact on the mental health of these individuals [51]. In line with this, Carrizo (2014) highlighted that the professionals interviewed related the term gender to violence against women and reproductive health [49].

#### 2.2.2. Training and Knowledge of Professionals

Four studies explored, through in-depth interviews or focus groups, the training and knowledge of the professionals [50,53,55,57] stressing the importance of training in diversity and multiculturalism to reduce structural barriers in the health services. In addition, the lack of training received during their educational stage was one of the main complaints of the professionals studied.

Professionals’ lack of training was suggested as one of the main barriers to accessing the health system for transgender and gender diverse people [50]. The participants interviewed by Torres et al. (2015) stressed that the need for training does not only include medical personnel, but must include all the disciplines related to the care of trans people, such as social workers, hospital workers, and nursing staff [53].

On the other hand, the study of Salpietro et al. (2019) underlined that, given the small number of professionals with adequate training, there may be a need for specialized clinical supervisors who know the different challenges that trans people face in family, internal, and socio-political environments [57].

#### 2.2.3. Accessibility of the Health System for Trans People

Four of the selected articles addressed the accessibility of the health system for trans people [49,50,52,53]. All the included research agreed that the greatest difficulties they experience in accessing the health system are due to structural, interpersonal, and individual barriers. As mentioned in the previous section, the lack of training of health personnel was identified as one of the main barriers in the health system and, at the same time, the most easily modifiable [50,52,53].

Carrizo (2014) found more obstacles than facilitators in access to the health system for trans people. Among the obstacles suggested were the scarce use of mental health facilities by these people, difficulties in knowing how to refer to these people according to their identity, the binary assignment of gender, professionals’ presumption of heterosexuality, and their attribution of sexual dissidence to illness [49]. Torres et al. (2015) added to these the difficulties for children or adolescents to know about the health system, and the need to improve the clinical environments by providing services such as inclusive bathrooms and information about the resources available to trans people [53]. Both Torres et al. (2015) and Tishelman et al. (2019) highlighted as a structural barrier the cost of access to hormonal treatment and fertility preservation in the USA [52,53].

#### 2.2.4. Therapeutic Tools When Working with Trans People

Four studies explored therapeutic tools that favor working with trans people [48,56,57,58]. Some authors highlighted the importance of addressing stigma in therapy as a useful tool to help patients develop connections and relationships with surrounding communities [48,57,58]. They also stressed the importance of individualized treatment, appropriate to the needs of each individual, and the role of the health professional to act as a defender of human rights, which are frequently violated for this group. Along this line, Acosta et al. (2019) and Salpietro et al. (2019) indicated the importance of the therapeutic alliance with patients, paying attention to aspects such as the proper use of pronouns, and empowering the patient directly as a source of training and familiarization with terms related to gender identity [48,57].

On another hand, the professionals interviewed by Heng et al. (2019) underlined the essential role of the clinician to provide support during the different difficulties experienced by users, as opposed to the traditional role of help in decision making [56].

## 3. Discussion

The main objective of this review was to evaluate the research conducted on the attitudes and beliefs of mental health professionals towards the transgender community. Likewise, we proposed to study the factors related to the attitudes and beliefs of these professionals, with special attention to social, formative, and cultural aspects.

The results of our review indicated that there is a trend of positive attitudes among mental health professionals towards transgender people compared to the general population [32,34,36,37,38,45,50]. Such positive or favorable attitudes are related, according to our study, to social or cultural factors such as belonging to a racial or sexual minority, liberal political ideologies, or greater professional experience [8,33,42,43]. Likewise, some authors highlighted the importance of the professionals’ own beliefs, such that positive attitudes toward transgender people were related in those professionals who attributed causes of a psychosocial nature to gender identity [42,54].

However, negative attitudes of mental health professionals towards the trans and gender diverse community were also observed among the reviewed studies [33,36,43,46], which may detrimentally influence the treatment received by trans individuals seeking assistance in mental health services, according to different authors [3,25,32,37]. In this sense, knowledge deficits, lack of cultural sensitivity, and discrimination by professionals have been pointed out as common barriers suffered by the trans and gender diverse community in their contact with mental health services [60].

The present study has linked negative attitudes of mental health professionals to social and cultural factors, such as professionals’ personal biases, the presence of gender ideology, conservative political affinity, or religious beliefs [33,42,43,54,55]. The evidence in this regard should be considered with caution, as it comes from a limited number of observational studies with variable methodological quality. However, it does confirm the results of recent studies linking the religious beliefs and conservative political positions of psychiatrists, psychologists, and nurses with unfavorable attitudes towards transgender people [61].

Although some authors included demographic variables such as age or gender in their studies [17,33,43], none of them reported sufficient evidence to claim that they influenced the attitudes of the professionals surveyed. In fact, there are studies that suggest that, if controlled for in comparison with other variables such as cultural bias, empathy, or personal contact with trans persons, they would cease to be significant, and they express the need for studies to determine what might be mediating the relationship of gender and age with attitude [25].

Based on the results of our review, we propose reducing negative attitudes among mental health professionals towards the transgender population by (a) prioritizing the therapeutic alliance between the professional and the patient [44]; (b) the inclusion of therapists belonging to sexual minorities in multidisciplinary teams [5]; (c) promoting social support through support groups and interaction with peers, especially in rural and isolated areas [53], and (d) encouraging spaces for exchange and collaborative training among professionals and users [47]. The implementation of specific curricular content on gender and diversity in all stages of the training of mental health professionals is also considered interesting in this regard, due to its potential effect on reducing the negative attitudes of professionals [8,31,45,46,47,49,52,57].

From this review, it is also clear that mental health professionals do not routinely receive, in daily care practice, basic information on the application of affirmative responses in the care of TGD persons, which coincides with what has been pointed out by previous studies [62,63,64]. Even the APA has warned of the importance of mental health professionals enhancing their competence through training aimed at recognizing gender diversity, as well as encouraging personal reflection around their beliefs about gender diversity [2]. In this regard, we propose brief online refresher training due to its low cost, short duration, and easy distribution in different treatment centers [47].

Finally, it should be noted that some of the studies reviewed warn about the importance of the institutional role, both in reducing access barriers and in modifying the behavior of professionals, given their role and responsibility to promote a positive environment in work teams [46]. The guidelines set forth by the Standards of Care for the Health of Transgender and Gender Diverse People, Version 8, could be useful in this regard [1], to which should be added the importance of the role of managers of these services and their involvement in developing a transgender-positive organizational approach, ensuring policies, care practices, and environments that support gender expression, and modeling attitudes of respect and inclusion among mental health professionals [65].

## 4. Limitations and Strengths

To our knowledge, this review was the first one to examine the influence of sociodemographic variables on the attitudes towards trans people of mental health professionals, which includes all health personnel involved in the therapeutic process. In this sense, our work has also addressed fundamental aspects such as the influence of training on the professionals’ behaviors.

However, the present study presents important limitations that should be considered when interpreting the results. On the one hand, the limited methodological quality of many of the studies that addressed the attitudes of mental health professionals towards the trans population is noted, an aspect that was mitigated through risk-of-bias analysis.

Likewise, the high heterogeneity of the outcome measures analyzed in the quantitative studies included in the present study prevented the meta-analysis of the data, as recommended by PRISMA [26]. In this sense, we note that most of the studies reviewed were cross-sectional and had a limited sample.

On the other hand, there were several limitations of the collated data that warrant further discussion. First, the quantitative and qualitative data collected in this review related predominantly to a female, white, heterosexual, Western population, with studies conducted mostly in the USA. Although some data were collected from research conducted in non-Western regions, it is not possible to make a comparison between results due to the variability of measurement tools employed. This reduces the generalizability of the results obtained and requires caution in interpreting the results. The inability to ascertain the views of under-represented groups (non-white populations and sexual/ethnic minorities) may reflect a difficulty in recruiting these populations or methodological flaws in the design of the studies reviewed.

## 5. Implications for Practice and Research

The lack of training and ignorance of a large part of the health community are noteworthy, as is the need to integrate specific content on gender and sexual minorities into the university curricula of the socio-health professions.

Regarding the implications for research, most of the quantitative studies reviewed reflect investigations in the presence of negative or positive attitudes and the demographic variables that influence them. Instead, very few provided the components that build a positive or negative attitude. While several of the authors included in this review have attempted to investigate associations between demographic variables and attitudes, the impact on the interaction between trans and gender diverse people and mental health professionals is neglected and requires further investigation. In addition, it would be interesting to use homogeneous valid instruments and outcome measures that allow meta-analysis of the results. On the other hand, we recommend expanding the studies that use qualitative research methodology to examine in more depth the results derived from the interventions, as well as considering the use of mixed methods to achieve a broader and deeper vision of the phenomenon under study. Thus, more studies are needed to try to understand what interventions are indispensable to reduce the differences perceived by trans people when demanding psychiatric or psychological treatment in the Public Health System, as well as the impact these actions would have on the perceptions of these differences by trans people.

## 6. Conclusions

The reviewed studies suggest that mental health professionals have mostly positive attitudes towards transgender and gender diverse people. Such attitudes may be subject to the psychosocial and individual characteristics of the therapists, especially previous experiences with transgender people, personal concerns, and a therapist’s own belonging to a minority group.

Situations of negative attitudes of mental health professionals towards transgender and gender diverse people are related to social and cultural factors, such as the personal biases of the professionals, the presence of gender ideology, conservative political affinity or religious beliefs.

There is insufficient evidence to confirm that demographic variables, such as age or gender, influence the attitudes of mental health professionals towards transgender people.

Finally, the importance of training mental health professionals, and the institutional role in promoting positive and open attitudes towards transgender and gender-diverse people, are highlighted.

## Figures and Tables

**Figure 1 ijerph-20-06495-f001:**
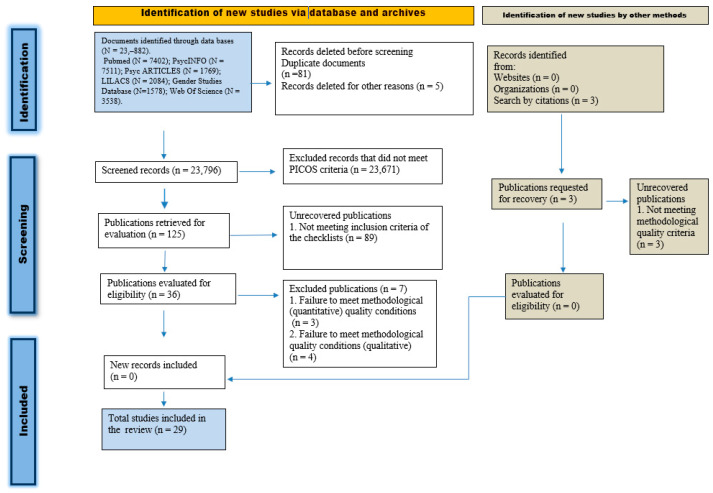
PRISMA flow diagram for selected references [26].

**Table 1 ijerph-20-06495-t001:** Quantitative studies included in this systematic review.

Study/Country/Design	Objective	Sample	Outcome Measures/Instruments	Main Findings	Conclusions
Riggs D.W., Bartholomaeus C. (2016)/Australia Design: cross-sectional, comparative [8]	-To identify correlations between demographic variables identified in previous research (age, gender, religion, experience, etc.) with measures of clinical knowledge, attitudes, and confidentiality when working with trans people.	N = 304 Gender: Female: 78.3%; Male: 21.7% Age: M = 44.64 (SD = 11.85)	-Training and experience (ad hoc). -Adapted version of the Counselor Attitude Toward Transgender Scale (CATTS) and Attitudes Towards Transgender Individuals Scale (ATTIS) -Confidence in Working with Trans Clients Measure (CWTCM).	-Experience and training in trans: Psychiatrists and nurses were more experienced than other professionals. -Clinical knowledge: Psychiatrists had less knowledge than psychologists, social workers, and counselors. -Comfort: It was related to clinical knowledge.	-Being a woman, training, and previous experiences with trans people predicted higher levels of clinical expertise. -Psychiatrists reported poorer levels of clinical knowledge, despite having worked more with trans people than the other groups. -Experience and training were predictors of self-confidence. -The importance of training for professionals who have worked or may work with trans people at some point.
Whitman C.N., Han H. (2016)/USA Design: mixed [12]	To evaluate formative experiences, understanding of terms, and competencies concerning trans people among mental health personnel at different levels of training.	N = 53 Gender: Female: 39 (73.6%); Male: 9 (17%); Trans: 1 (1.9%); Queer: 1 (1.9%); Indeterminate: 3 (5.7%) Age: 18–72	-Gender Identity Counselor Competency Scale (GICCS). -The TGNC Knowledge Assessment (KA), prepared for the study. -The Social Desirability Questionnaire (SDQ). -Free response clinical vignettes.	-Professionals who were more comfortable and interested in working with trans people had more training and education in this matter. 88.7% of the participants did not consider being trans a mental pathology. -Clinicians in training knew gender vocabulary better than those who had already graduated.	-The professionals who strive to acquire multicultural competencies and be aware of their limitations were less likely to perpetuate either explicit or implicit stigma.
Ali N., Feiseh W., Ericson J. (2016)/Canada Design: cross-sectional [17]	To investigate the attitudes of psychiatry residents and psychiatrists towards trans people.	N = 74 Gender: Female 31 (41.9%); Male: 42 (56,8%) Age: 20–29 70–79	-Genderism and Transphobia Scale (GTS).	-Psychiatrists and residents presented more positive attitudes than the general population. -A lower trend in scores among child psychiatrists (not statistically significant).	-Despite not receiving specialized training, psychiatrists and residents appeared to have a tolerant attitude towards trans people. -Psychiatrists may be influenced similarly to other medical professions.
Cherabie J., Nilsen K. (2018)/USA Design: quasi-experimental [31]	-A training course was carried out for teachers, students, and medical residents on the health of transsexuals.	N = 163 N/A N/A	Author elaboration: -Beliefs. -Attitudes. -Comfort. -Knowledge.	It does not provide specific results for mental health. -Knowledge: Significant improvement after the intervention: (t [264] = 12.83, *p* < 0.0001, 95% IC [1.17, 1.59]. -Attitudes: Significant improvement after the intervention (t [271] = 16.90, *p* < 0.0001, 95% IC [1.33, 1.67].	Exposure to training has the potential to influence attitudes, comfort when dealing with transgender people, and knowledge about this collective.
Franzini L.R., Casinelli D. (1986)/USA Design: cross-sectional [32]	To determine the knowledge and attitudes towards trans patients and sex reassignment surgery in a sample of professionals. To compare the results with a previous study.	N = 225 Gender: Female 17%; Male: 83%. Age: 30–93	-Knowledge/questionnaire of 24 items; author elaboration. -Attitudes towards trans patients and reassignment surgery/questionnaire of 30 items; author elaboration. -Replication of the questionnaire of the previous study of Green et al.	-The different medical specialties analyzed had a similar level of knowledge about trans. -Psychologists had more liberal attitudes toward trans people than medical professionals. Mean Favorability Rating: Psychologist: 80.63; Psychiatrists: 73.28.	-All participants had similar knowledge. -General practitioners were the most conservative, whereas clinical psychologists had the most favorable attitudes among the respondents.
Johnson L., Federman E.J. (2014)/USA Design: cross-sectional [33]	To determine the training, experience with, and attitudes towards LGBT. To measure the practices of a national sample of psychologists of the public administration and to determine whether they vary by region and age of the professionals.	N = 384 Gender: Female: 65.4%; Male: 34.4%; Male Trans: 0.3%. Age: M = 45	-Current practices. -Attitudes and Knowledge. -Self-referenced competence. -Interest in training.	-Most participants had received little or no training about transgender people during their university stage. -Training and formation in sexual orientation were significantly higher than in trans issues. -92% of participants do not routinely ask about gender identity. -More than 35% of the psychologists said they had never seen a trans client. -Young psychologists had more training and more positive attitudes.	-The need for more specific training in LGBT issues in the health field. -The therapist’s identity can impact their multicultural or self-perceived competence.
Kawano T., Cruz R.F., Tan X. (2018)/USA Design: mixed [34]	To study the personal and professional attitudes of therapists from the movement towards gay, lesbian, bisexual, trans, queer, intersex, or non-congruent people.	N = 361 Gender: Female: 91%; Male: 9% Age: 18-58	-LGBTQI attitudes/adapted from a previous questionnaire by Whitehead-Pleaux et al. (2013) for drama therapists. -Social desirability.	-Most of them identified at least one social, political, or legal issue of interest to LGBTQI communities. -The use of toilets seemed to reflect a broader problem of gender binary infrastructure. -The main socio-political problem identified was discrimination (31%).	-The results highlight therapists’ good intentions towards LGBTQI communities and actively explore ways to achieve this. -The lack of training to work with this population was evident, highlighting ignorance of specific issues of the LGBTQI community such as the use of language, documentation, and legal rights.
Naal H., Abboud S., et al. (2019)/Lebanon Design: cross-sectional [35]	To examine attitudes and behavior towards LGBT people in a sample of health personnel.	N = 141 Gender: Female: 39.7%. Male: 60.3% Age: M = 39.4, SD = 9.96.	-Attitudes and behaviors, a scale developed by the researchers.	-Mental health professionals were more likely to interact with LGBT people than other health workers. They were less likely to think it is unnatural to identify with a gender different from the biological sex. They were significantly more willing to refer to trans people according to the gender they identified with.	-Mental health professionals were more likely to show favorable attitudes.
Willoughby B., Hill D., González C.S., et al. (2010)/USA Design: cross-sectional [36]	To examine negative attitudes toward trans people among mental health professionals. To evaluate the impact of personal contact with trans people on attitudes towards gender non conformity. To examine the confidence, validity, and structural factor of the GTS scale among university students. -To create a comprehensive model predicting gender and transphobia.	N = 88 Gender: 77%; Male: 22% Age: 20–57	-Genderism and Transphobia Scale (GTS). -Knowledge About Transgender Individuals.	48% of the participants scored 8/10 in the questionnaire on knowledge about trans people. -Scores on the GTS scale: M = 57.8, SD = 21.0, indicating that most of the surveyed staff had low levels of transphobia. -Professionals who had received training in gender and sexuality had a lower level of transphobia.	Mental health professionals have relatively lower levels of transphobia and gender stereotypes than the general population. Knowledge and training in sexuality can reduce negative attitudes.
Bolding D.J., Acosta A. (2022)/USA Design: cross-sectional [37]	To explore occupational therapy practitioners’ self-reported knowledge about, clinical preparedness for, and attitudes toward working with LGBT clients.	N = 48 Gender: Female: 92%; Male: 6%; Gender minority: 2% Age: 20–29: 17%; 30–39: 36%; 40–49: 20%; 50–59: 20%; ≥60: 7%	-Demographic characteristics, education and workplace knowledge, clinical. -Lesbian, Gay, Bisexual, and Transgender Development of Clinical Skills Scale (LGBT–DOCSS).	-Degree level, continuing education, practice setting, minority sexual orientation, and having a close friend or family member who identifies as SGM were associated with higher mean scores on the LGBT–DOCSS. -Higher religiosity and frequency of religious practice were associated with lower scores on knowledge and attitudinal awareness.	A basic level of continuing education can improve occupational therapy practitioners’ knowledge of and skills for working with LGBT populations.
Vann D.M., Riggs D.W. (2021)/Australia Design: quasi-experimental [38]	To evaluate the utility of brief online training for facilitating mental health professionals’ perceived knowledge, confidence, and comfort in working with non-binary clients, and building positive attitudes.	N = 38 Gender: Female: 32 (84.2%); Male: 6 (16.8%) Age: M = 32.45 SD = 11.35	-Demographic and professional variables. -Perceived knowledge about non-binary people. -Attitudes towards the Inclusion of Transgender Women in Domestic Violence Services Scale, an adapted version of Riggs et al. (2016).	-Follow-up measures completed one-week post-intervention indicated engagement in the training may lead to increased perceived knowledge, confidence, and comfort. -Attitudes were positive at baseline and were not significantly different at follow-up.	Potential utility of a brief online training module for building positive attitudes, perceived knowledge, confidence, and comfort of mental health practitioners for working with on-binary clients.
Powell H.A., Cochr B.N. (2021)/USA Design: cross-sectional [39]	To explore the relationships between mental health care providers’ transphobia, gender minority-knowledge, and hypothetical treatment decisions when working with gender-minority clients	N = 107 Gender: Female: 56 (70%); Male: 21 (26.3%); Nonbinary: 2 (2.5%); Trans man: 1 (1.2%) Age: M =36.4; SD = 11.8	-Knowledge of gender minorities. -Genderism and Transphobia Scale (GTS). -Treatment decisions: Six vignettes elaborated by the researchers.	-Transphobia negatively predicted knowledge, *F*(1,85 = 24.16), *p* = 0.02, R^2^ = 0.22. -Transphobia was significantly predictive of treatment decisions in the sample, *F*(1, 76) = 33.66, *p* < 0.01, R^2^ = 0.31. -Higher transphobia scores were associated with a wider discrepancy in treatment decisions between transgender clients.	Stronger transphobic attitudes may be linked to both providers’ dearth of knowledge about gender-minority individuals and different treatment decisions regarding gender minority clients.
Francia-Martínez M., Esteban C., Lespier Z. (2017)/Puerto Rico Design: cross-sectional [40]	-To examine the attitudes towards, knowledge about, and social distance from the trans community in a sample of psychologists and graduate students of Psychology.	N = 233 Gender: Female: 77%; Male: 21.9%; Trans: 0.5%; Queer: 0.5% Age: 22–84	-Attitudes and knowledge towards transgender and transsexual people (AC-TT), developed for the study. -Social distance from trans people (DS-T scale), developed for the study. -Questionnaire on Diagnoses of the Trans Community (C-DT); author elaboration.	-14.9% showed moderate prejudice towards the transgender community, and 19.8% towards the transsexual community. -21.2% feel anxiety when they first see a transgender client. -Social distance: Approximately 50% of the sample reported that they would accept trans people as friends, but would not live under the same roof. -63.4% would remove the diagnoses for transgender and/or transsexual people from the DSM.	-The results in competencies and knowledge of the participants suggest that a significant number of therapists are unaware of important aspects of the problems of this group.
Gaspodini I.B., Falcke D. (2018)/Brazil Design: cross-sectional [41]	To study the relationship between prejudices and beliefs about the nature of sexual diversity in Brazilian psychologists.	N = 497 Gender: Female: 396 (79.7%); Male: 97 (19.5%) Age: 22–69	-Scale of Crenças on a Natureza da Homossexualidade modified. -Scale of Prejudice against Sexual and Gender Diversity, Revised (EPDSG-R).	-The mean value of extreme prejudice against sexual and gender diversity was 1.46 (SD = 0.44). -Heterosexual participants had significantly higher average prejudice; U = 13,120.50, *p* < 0.01. -Beliefs about LGTBQ: the psychosocial origin obtained the highest score.	-Extreme prejudice among professionals was medium-low. -Heterosexual psychologists expressed more prejudice than non-heterosexuals.
Dispenza F., O’Hara C. (2016)/USA Design: cross-sectional [42]	To determine the relationship between sociodemographic variables and competencies to treat trans and gender non-congruent people.	N = 113 Gender: Female: 78%; Male: 22% Age: 18–58	-Sociodemographic and personal experience. -Gender Identity Counselor Competency Scale (GICCS). -Social Desirability Scale-17 (SDS-17).	Variables that contributed significantly to improve the professionals’ competence: (a) To identify as a sexual minority. (b) Professional experience.	-Variables related to the identity of professionals contribute to their knowledge, tools, and attitudes. -The importance of including people belonging to sexual minorities in professional teams to promote an affirmative environment towards trans people.
Vijay A., Earnsaw V., et al. (2018)/Malaysia Design: cross-sectional [43]	-To explore factors associated with intentions to discriminate against trans people.	N = 436 Gender: Female: 206 (47.2%); Male: 230 (52,8%) Age: M = 34.7	-Modified version of the Stein and Li multidimensional HIV Stigma Scale. -Constructs related to stigma: prejudice towards trans, internalized shame, beliefs, stereotypes.	-The mean score for “intentions to discriminate” was M = 1.81 (SD = 0.58). -Psychiatrists presented less intention to discriminate than other medical specialties. -Prejudice, internalized shame, fear, and stereotypes correlated positively with intention to discriminate.	-The importance of recognizing the responsibility of professionals in the care of trans people. -To achieve equality of care is a multifactorial process that requires advances in education and training but also legal and political advances.
Craig S.L., et al. (2015)/Canada Design: cross-sectional (pre-experimental) [44]	A 3 h training was conducted by community-based organizations, including information on terms, risks and resilience, discrimination, and specific tools and strategies for work with sexual minority youth. To analyze a community educational intervention, improve the knowledge and tools of public professionals in the care of young people of sexual minorities.	N = 2850 (327 health personnel) Gender: Female: 1796 (67.4%); Male: 792 (29.7%) Age: 18–66	-The Self-administered CBEI Survey (author elaboration). They measured the influence of training on participants, and included questions about knowledge of and attitudes toward LGTQI.	-Participants valued the experience positively. -Behavioral intention: 79.5% of the participants intended to carry out at least one action to improve the lives of young people of sexual minorities.	-Training is useful to improve professionals’ attitudes. -Participants stated that their knowledge had improved, which was linked to the decrease in negative attitudes towards this group.
Lelutiu-Weinberger C., Pachankis J.E. (2017)/Romania Design: quasi-experimental [45]	-To evaluate the impact of a pilot training about LGBT people among mental health professionals in their knowledge, perception of their clinical skills, and to reduce their negative attitudes.	N = 40 Gender: Female: 90%; Male: 10% Age: 25->50	-Previous training about LGBT. -Attitudes, skills, and knowledge/adapted original scales to include trans people. -Modern Homonegativity Scale modified.	-Participants felt more comfortable when dealing with LGTBQI people. Homo-negativity and trans-negativity were significantly reduced.	Reducing negative attitudes towards the LGTBQI community requires a multidimensional intervention that includes modifying attitudes and practices among individuals, groups, and institutions.
Mizock L., Hopwood R., et al. (2017)/USA Design: pre/post-experimental [46]	To evaluate the effectiveness of transphobia reduction in the training “The Transgender Awareness Webinar” adapted for mental health professionals.	N = 158 Gender: Female: 124 (78.5); Male: 27 (17.1%); Trans: 7 (4.4%) Age: 22–67	-Experience and sociodemographic information. -The Transphobia Scale.	-Participants reduced their scores on transphobic attitudes after the training (M = 6.29, SD = 0.77); M = 5.97, SD = 0.84, t(139) = −11.99, *p* < 0.001. -They found no significant differences in the participants who had received previous training or had more experience with trans people.	-Training is potentially useful to train mental health professionals.

**Table 2 ijerph-20-06495-t002:** Qualitative studies included in this systematic review.

Author/Country	Objective	Sample	Sources of information	Analytic Method	Categories	Conclusions
Acosta W., Qayyum Z., et al. (2019)/USA [48]	To deepen the experience of trans adolescents (age: 13–17) and psychiatrists of a psychiatric hospital unit in the USA.	*N =* 18 Gender: N/A Age: N/A	Individual interview.	Thematic analysis.	-Identification of patients with a non-congruent gender identity. -Therapeutic alliance between professionals and these patients. -To understand the complexities of gender identity.	Patients and professionals, in general, described a respectful hospitalization environment. The main factors noted were efforts by professionals to respect gender identities and acknowledge their mistakes.
Carrizo Villalobos C. (2014)/Argentina [49]	To identify obstacles to and facilitators of accessibility to mental health services. To describe the characteristics of mental health policies. To characterize professionals’ meanings and practices in gender diversity.	*N =* 8 Gender: N/A Age: N/A	In-depth interview.	Grounded theory.	-Accessibility. -Discrimination: Recognition and assessment of discrimination in primary and secondary settings. -Gender diversity: Meanings of sexuality and gender. -Mental Health Services. -Public policies relationship and socio-political organizations.	More obstacles than facilitators of psycho-sociocultural accessibility were identified. Highlighting: differences in representations of sexuality and gender, professionals relate transsexualism with prostitution and/or disability, lack of respect for the identity or orientation of trans people, and recurrent situations of discrimination against LGTBQI people.
Clark K.A., White Hughto J.M., et al. (2017)/USA [50]	To examine attitudes, knowledge, and experiences of prison health professionals toward incarcerated trans people.	*N =* 20 Gender: Females omen: 90%; Males: 10% Age: N/A	Semi-structured individual interview.	Strauss and Corbin’s Grounded theory and thematic analysis.	-Structural barriers: Little training, restrictive policies, budgets, etc. -Interpersonal barriers: Interactions and conflicts between safety and health personnel. -Individual barriers: Lack of clinical training derived from personal trajectories and lack of knowledge and experience.	The interviewed professionals stated that trans people do not receive adequate assistance while incarcerated. The interviews showed bias and discrimination among security and health personnel.
Lefkowitz A.R.F, Mannell J. (2017)/United Kingdom [51]	To examine the attitudes of sexual health providers towards trans youth in England and the implications for their clinical practice.	*N =* 20 Gender: N/A Age: N/A	Semi-structured individual interview.	Attride-Stirling’s thematic analysis.	-Binary representation of trans people. -Transgender as homosexuals. -Uncertain bodies. -Unstable mental states. -Too young to decide.	Further training on trans health is recommended for service providers, highlighting the need for specific guidance in the content of this training to foster open debates about transgender experiences, sexualities, and bodies.
Tishelman A.C., Sutter M.E., et al. (2019)/USA [52]	To examine the perspectives of health care providers in behaviors and barriers related to fertility advice, fertility preservation, and family building in adult and youth transgender patients.	*N =* 110 Gender: N/A Age: N/A	Written response to four open-ended questions.	Inductive analysis of the content and method of constant comparison.	-Advice on fertility and contraceptive methods. -Perceptions of the role of responsibility. -Perceptions of the parental role in decision-making. -Barriers in access to fertility preservation techniques.	They highlight the need for specific training in fertility and resources for professionals. It is necessary to differentiate the role of mental health professionals from the rest of the health personnel who serve trans youth.
Torres C.G., Renfrew M., et al. (2015)/USA [53]	To understand the relationships between the personal characteristics and care needs of trans youth (age: 13–21) that contribute to increasing their resilience.	*N =* 11 Gender: N/A Age: N/A	Individual interviews.	Strauss and Corbin’s grounded theory.	-Resilience of trans youth. -Lack of access to services. -The essential role of social support. -The complexities of the health system. -Education and training for all the staff.	Professionals recognize multiple barriers and challenges in caring for trans youth. However, they also identify the resilience manifested by many young people.
Whitehead J.C., Thomas J., et al. (2012)/USA [54]	They interviewed various professionals who described themselves as “trans-friendly”, focusing on those cases in which clients had been denied access to the modification of their bodies.	*N =* 35 Gender: N/A Age: 45–50 (most of them)	Semi-structured interview.	Grounded theory.	-The role of the clinician and the diagnostic process. -Therapist’s opinion and use of the DSM manual. -Professional belief about gender acquisition. -Crucial cases in which the therapist denied, delayed the diagnosis, or misdiagnosed.	Despite having selected a respectful sample of clinicians, they had varied views on gender identity and did not share criteria about when to deny access to body modifications.
Gaspodini I.B., Falcke D. (2018) /Brazil [55]	To study how psychology professionals experience issues of sexual diversity.	*N =* 14 Gender: Females: 100% Age: 24–60	3 focus groups in 3 regions of the State of Rio Grande do Sul.	Thematic analysis.	-Beliefs and attitudes of psychologists. -Clinical experience. -Clinical training.	Professionals who carry out depathologizing practices are motivated by three factors: (1) belief in the psychosocial nature of diversity; (2) concern about the reproduction of stereotypes; (3) clinical training, self-knowledge, and interpersonal contact with LGTBQI people.
Heng A., Heal C., et al. (2019)/Australia [56]	To explore the perspectives and experiences of trans clinicians and healthcare users in North Queensland, Australia.	*N =* 8 Gender: Females: 62.5%; Males=: 5%; Non-binary: 12.5% Age: 24–69	Semi-structured individual interview.	Inductive content analysis.	-Community attitudes and support in the region. -Trans health “is not just a matter of hormones”. -Clinicians who “went above and beyond” to help. -Learning together.	The authors recommend the formation of support groups, especially in rural or isolated areas, as well as collaborative and holistic training that includes interaction between users and clinicians.
Salpietro L., Auslos C., et al. (2019)/USA [57]	To examine the experiences of cisgender therapists, highlighting their positive and negative experiences, education, clinical training, therapeutic alliance, and values when working with trans people.	*N =* 12 Gender: Females: 83.3%; Males: 16.6% Age: 26–65	Individual interviews.	Phenomenological analysis.	-Challenges in treatment. -Learning experiences in cisgender therapists. -Basic training. -Therapeutic tools.	Many trained therapists are not prepared to work with trans people, highlighting the importance of education in diversity and multiculturalism. Another important issue was the lack of qualified supervisors to oversee the work with trans people.
Holt N.R, Hope D.A. (2019)/USA [58]	To describe services provided by mental health professionals who self-describe themselves as close to the trans community in a rural area of the USA.	N = 10 Gender: Females: 70%; Males: 30% Age: 44–76	Semi-structured individual interview.	Weiss approximation method (classification, local and inclusive integration).	-Work topics in therapy. -Considerations when working with trans people.	Trans people living in underdeveloped areas face significant inequalities and barriers to access the health system. Although there are some respectful professionals in these areas, we still hope for evidence-based care in these communities, which present large inequalities.

## Data Availability

The methods used were pre-specified and documented in a protocol available online (https://www.crd.york.ac.uk/prospero/display_record.php?ID=CRD42022177761 (accessed on 22 July 2023)).

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
