# Peer review of "Attitudes and Beliefs of Mental Health Professionals towards Trans People: A Systematic Review of the Literature"

_ijerph, 2023, doi:10.3390/ijerph20156495_

Round 1

Reviewer 1 Report (Previous Reviewer 1)

First of all, thanks for taking the remarks of this reviewer seriously.

In general  the current manuscript has no structural problems or flaws. However, there remain some minor problems which deserve improvement:

-Line 37-39: "These barriers, which range from individual obstacles of the professionals themselves to organizational barriers, cause difficulties both in access to socio-health systems and the care itself in health and disease processes"  Given that no definition is given of "socio-health systems", it remains unclear what socio-health systems are. In consequence, this reviewer suggests the following reformulation: cause difficulties in access to health  care systems.

-Line 39-42: "In this respect, the study conducted  by Lambda Legal (2010) with lesbian, gay, bisexual, and transgender (LGBT) participants in the United States found that 20.9% of those identified as transgender person 41 had suffered insults from health workers, and 20.3% said they felt blamed for their 42 health problems [11]." Why are you referring to only one example from the USA? What about all the other countries in the world? Especially since the USA are not a transgender and gender diverse friendly country! Please expand your examples. 

-Line 58-59: "Most span the entire LGBT spectrum [18–25] and tend to focus exclusively on  the medical community" Insert after  the medical community a reference, which has shown empirically that there is a focus exclusively on  the medical community.

-Line 59-60: "Furthermore, we must highlight the great heterogeneity found regarding the conceptualisation of the term among the different instruments used to measure attitudes on health professionals," It is not clear to which concept/variable  the conceptualisation of the term  is referring. Please clarify; for example by inserting the name of the term after the word term.

-Line 67: "Mental Health professionals", insert defined as psychologists, .....

-Line 76: “What attitudes do mental health service  professionals have towards trans people?” In this question there is no reference to beliefs. It is given in consideration to delete recence to beliefs in the whole manuscript and to limit the text to attitudes. 

-Line 80: "The methods used were pre-specified and documented in a protocol available online." Insert here a link to this protocol.

-Line 89: "No temporal or language restrictions were established, given the limitation of similar studies" . I do not understand what the authors mean by "given the limitation of similar studies". Please clarify.

-Line 101: " can present" ; replace by used.

-Line 120: "evaluation scale". Please insert the content or items of the evaluation scale (and not only the scoring).

-Line 145: "were grouped into common themes". Please insert how and by whom this was done.

-Line 195-196 "On another hand, Johnson and Federman (2014) found more positive attitudes  among young therapists, suggesting a likely greater impact of early training on sexual diversity and the long-term impact on their professional career" I do no understand this suggestion/interpretation. As an example the following alternative: this is a result of having grown up in a more transfriendly society and has nothing to do with their training. So, please, check if your suggestion is appropriate methodologically given the design used bv Johnson and Federman (2014).

-Line 209: "politically progressive states". Please define. Now the meaning is implicit and confusing (as if every body agrees on what progressive is). And the same applies to "a politically conservative state," (Line 215)

-Line 316: "that trans people face in family, internal, and socio-political environment". Is it not more correct to write environments?

-Line 448: "to use homogeneous instruments and outcome measures". Suggestion: insert and valid afer homogeneous

Author Response

Response to reviewer 1

1) Reviewer 1 comments

-Line 37-39: "These barriers, which range from individual obstacles of the professionals themselves to organizational barriers, cause difficulties both in access to socio-health systems and the care itself in health and disease processes"  Given that no definition is given of "socio-health systems", it remains unclear what socio-health systems are. In consequence, this reviewer suggests the following reformulation: cause difficulties in access to health  care systems.

AUTHORS: The suggestion is gratefully acknowledged and incorporated into the manuscript.

2) Reviewer 1 comments

-Line 39-42: "In this respect, the study conducted  by Lambda Legal (2010) with lesbian, gay, bisexual, and transgender (LGBT) participants in the United States found that 20.9% of those identified as transgender person 41 had suffered insults from health workers, and 20.3% said they felt blamed for their 42 health problems [11]." Why are you referring to only one example from the USA? What about all the other countries in the world? Especially since the USA are not a transgender and gender diverse friendly country! Please expand your examples. 

AUTHORS: We are grateful for the reviewer's very justified indication.  Instead of adding more examples, in order not to increase the length of the article too much, and taking into account the word limitation from the journal, we replaced the citation of Lambda Legal (2010) with an umbrella review, based on systematic reviews and meta-analyses and thus covering the entire international landscape (Zeeman et al., 2018).

3) Reviewer 1 comments

-Line 58-59: "Most span the entire LGBT spectrum [18–25] and tend to focus exclusively on  the medical community" Insert after  the medical community a reference, which has shown empirically that there is a focus exclusively on  the medical community.

AUTHORS: The requested references are added. Classically, studies have focused on all health personnel without differentiating exclusively on mental health personnel.

4) Reviewer 1 comments

-Line 59-60: "Furthermore, we must highlight the great heterogeneity found regarding the conceptualisation of the term among the different instruments used to measure attitudes on health professionals," It is not clear to which concept/variable  the conceptualisation of the term  is referring. Please clarify; for example by inserting the name of the term after the word term.

AUTHORS: The term referred to (attitudes) in the paragraph noted by the reviewer is clarified.

5) Reviewer 1 comments

-Line 67: "Mental Health professionals", insert defined as psychologists, .....

AUTHORS: We are grateful for the reviewer's recommendation. In the revised manuscript, the specifications suggested by the reviewer in the paragraph above have been added.

6) Reviewer 1 comments

-Line 76: “What attitudes do mental health service  professionals have towards trans people?” In this question there is no reference to beliefs. It is given in consideration to delete recence to beliefs in the whole manuscript and to limit the text to attitudes. 

AUTHORS: We appreciate the reviewer's suggestion. However, both attitudes and beliefs are referred to throughout the article. In fact, it is a term that is included in the search strategy, and in some selected articles, beliefs are mentioned as much as attitudes. Although, as the introduction makes clear, they are not identical terms, many researchers have used them interchangeably. Consequently, we believe it is more appropriate to include the term within the PICO question, rather than to remove it from the text.

7) Reviewer 1 comments

-Line 80: "The methods used were pre-specified and documented in a protocol available online." Insert here a link to this protocol.

AUTHORS: The link to the protocol is inserted, as suggested by the reviewer.

8) Reviewer 1 comments

-Line 89: "No temporal or language restrictions were established, given the limitation of similar studies" . I do not understand what the authors mean by "given the limitation of similar studies". Please clarify.

AUTHORS: After reviewing the expression, we believe that it is misleading, probably aggravated by the translation. The meaning of this expression is that they did not set limits on the search strategy in order to make it as sensitive as possible, given the difficulty of finding homogeneous studies addressing the subject of our systematic review. After consultation with the rest of the research team, we decided to remove it from the text.

9) Reviewer 1 comments

-Line 101: " can present" ; replace by used.

AUTHORS: The replacement recommended by the reviewer is carried out.

10) Reviewer 1 comments

-Line 120: "evaluation scale". Please insert the content or items of the evaluation scale (and not only the scoring).

AUTHORS: Dear reviewer, in order to make the article more concise the full scale is not indicated in the article. These can be consulted online and are available in bibliography. However, the following explanatory sentence is added:

The scale contains ten items that assess both the research objectives, qualitative methodology and research method, and the reflexivity and ethical aspects of the study.

CASP is an internationally recognised and easily accessible scale, which anyone can consult in the literature.

11) Reviewer 1 comments

-Line 145: "were grouped into common themes". Please insert how and by whom this was done.

AUTHORS: The reviewer's suggestion is gratefully acknowledged, and the process followed in the revised manuscript is incorporated.

12) Reviewer 1 comments

-Line 195-196 "On another hand, Johnson and Federman (2014) found more positive attitudes  among young therapists, suggesting a likely greater impact of early training on sexual diversity and the long-term impact on their professional career" I do no understand this suggestion/interpretation. As an example the following alternative: this is a result of having grown up in a more transfriendly society and has nothing to do with their training. So, please, check if your suggestion is appropriate methodologically given the design used bv Johnson and Federman (2014).

AUTHORS: The cited article uses scales that assess the training, experience, practice, attitudes and knowledge of the participants to compare training and experience with attitudes and knowledge in different age groups. It is true that the socio-cultural characteristics of the groups are different but the article does not assess it or take this fact into account.

13) Reviewer 1 comments

-Line 209: "politically progressive states". Please define. Now the meaning is implicit and confusing (as if every body agrees on what progressive is). And the same applies to "a politically conservative state," (Line 215)

AUTHORS: The article divides participants from different U.S. states according to election results for the purpose of comparison. Clarification added in the text.

14) Reviewer 1 comments

-Line 316: "that trans people face in family, internal, and socio-political environment". Is it not more correct to write environments?

AUTHORS:  The reviewer's recommendation is accepted.

15) Reviewer 1 comments

-Line 448: "to use homogeneous instruments and outcome measures". Suggestion: insert and valid after homogeneous

AUTHORS:  The reviewer's recommendation is accepted.

Reviewer 2 Report (New Reviewer)

The study is well-structured, and the systematic review is well-executed. I have a few suggestions for improving the text:

1. In the first paragraph, on page 2, lines 49-52, it mentions "The difference between the two concepts is that...". However, it is unclear which "two concepts" are being referred to. The previous sentence explicitly discusses only one concept, that of "attitude": " "Various authors have defined attitudes or beliefs as a person's favorable or unfavorable evaluation of an object or social situation in terms of the consequences of their behavior toward it".

2. Line 228 - It is recommended to include the proper references for the four studies mentioned in the sentence: "Four cross-sectional studies addressed professionals' knowledge about consensual data on trans people."

3. Line 239 – It is recommended to add the references to "Two cross-sectional and one mixed study addressed professional competencies as an outcome measure."

4. In the discussion section, several significantly relevant points are highlighted regarding the attitudes of mental health professionals towards transgender people. However, the conclusion, consisting of only five lines, primarily focuses on the positive attitudes of mental health professionals towards transgender and gender diverse people, attributing these attitudes to the psychosocial and individual characteristics of the therapists, particularly their previous experiences with trans people, personal concerns, and their own membership of a minority group. This conclusion seems to downplay other equally important aspects raised in the analysis. It is suggested to provide a more comprehensive synthesis of the conclusions reached in the systematic analysis of the selected studies within the review.

Author Response

Response to reviewer 2

Reviewer 2 comments

  1. In the first paragraph, on page 2, lines 49-52, it mentions "The difference between the two concepts is that...". However, it is unclear which "two concepts" are being referred to. The previous sentence explicitly discusses only one concept, that of "attitude": " "Various authors have defined attitudes or beliefs as a person's favorable or unfavorable evaluation of an object or social situation in terms of the consequences of their behavior toward it".

AUTHORS: Gracias por la recomendación. En el párrafo mencionado nos referimos a las diferencias entre dos conceptos: actitudes y creencias, que algunos autores no distinguen claramente. Sin embargo, es posible que no se haya expresado con la suficiente claridad. Se modifica la frase para aclarar su significado, tal y como recomienda el revisor.

Reviewer 2 comments

  1. Line 228 - It is recommended to include the proper references for the four studies mentioned in the sentence: "Four cross-sectional studies addressed professionals' knowledge about consensual data on trans people."

AUTHORS: Thanks for the recommendation. Although they appear later, the citations are added at the end of the sentence to clarify the text, as recommended by the reviewer. In addition, an error is corrected (four studies are mentioned in this paragraph, when in fact there are three studies cited).

Reviewer 2 comments

  1. Line 239 – It is recommended to add the references to "Two cross-sectional and one mixed study addressed professional competencies as an outcome measure."

AUTHORS: Thanks for the recommendation. Although they appear later, the citations are added at the end of the sentence to clarify the text, as recommended by the reviewer.

Reviewer 2 comments

  1. In the discussion section, several significantly relevant points are highlighted regarding the attitudes of mental health professionals towards transgender people. However, the conclusion, consisting of only five lines, primarily focuses on the positive attitudes of mental health professionals towards transgender and gender diverse people, attributing these attitudes to the psychosocial and individual characteristics of the therapists, particularly their previous experiences with trans people, personal concerns, and their own membership of a minority group. This conclusion seems to downplay other equally important aspects raised in the analysis. It is suggested to provide a more comprehensive synthesis of the conclusions reached in the systematic analysis of the selected studies within the review.

AUTHORS: We appreciate the reviewer's advice. The conclusions had been limited for the adequacy of the text to the limitations of the magazine. In the revised manuscript, the conclusions are expanded in order to better capture the relevant points mentioned in the discussion.

This manuscript is a resubmission of an earlier submission. The following is a list of the peer review reports and author responses from that submission.

Round 1

Author Response

Regarding the manuscript: Attitudes and beliefs of Mental Health Professionals towards Trans People: a systematic review of the literature.

Manuscript ID:  ijerph-2237561

Dear reviewer, we deeply appreciate the review of the manuscript, which we believe will help to significantly improve the quality of our manuscript.

We will then proceed to respond as best we can to your suggestions, indications and requests for clarification. We have arranged our responses in the same order that they were listed in your review:

  • Reviewer: Throughout the manuscript the authors are to easy in making causal interpretations or conclusions about the determinants of attitudes and beliefs of mental health professionals towards trans people. Despite the fact that the authors signal that many of the reviewed studies have “a limited methodological quality”. Said in other words: the used designs do no allow for causal conclusions/inferences! How the can the authors make causal conclusions based on a “non-causal data base” of 29 studies?

Authors: We are grateful for the suggested indication. It is possible that we have improperly used some term or expression that could lead one to think that a causal inference is being made based on studies, in the majority of cases, of cross-sectional methodology and that therefore do not allow causality to be established. We proceed to revise the text to eliminate or modify these expressions and adapt them to the methodological limitations of the study.

  • Reviewer: In addition to this general methodological problem of the interpretation of the 29 included studies, the authors conclude (Line 426-429): “Attitudes towards a group cannot depend on individual experiences, so specific training in topics of gender diversity and minorities would be a key element for the improvement of care for trans people and for the reduction of inequalities in care and access, especially in those areas that are less developed or economically disadvantaged.”. However nearly no of the included studies studied in a controlled manner the effects of training on the attitudes and beliefs of mental health professionals towards trans people (see Table 1, in which nearly all studies have cross-sectional designs). So, once again the authors are to bold in their causal inferences. Please recognize more fully the methodological limitations of the included studies!

Authors: In line with the above suggestion, we proceed to revise the text in order to modify and adapt its wording to the methodological constraints.  We change the proposed paragraph to read as follows:

“The studies reviewed suggest positive attitudes of mental health professionals. However, these attitudes may be subject, to a large extent, on the psychosocial and individual characteristics of the therapists, especially previous experiences with trans people, personal concerns and the therapist's own membership of a minority group. Specific training in gender diversity issues is proposed as a potentially useful tool for improving the quality of care for trans people and for the reduction of inequalities in care and access.”

  • Reviewer: The concept of what a positive or negative attitude is theoretically is not clear in the manuscript. Is it a categorical or a dimensional variable? Please clarify. Clarify also what the meaning is of the conclusion “mental health professionals have positive attitudes towards trans people”. What is a positive attitude and how positive are these attitudes. In other words: expand the contextualization of the concepts attitudes and beliefs in the introduction! The same holds for the concept “liberal attitudes”. Used for example in Line 184; but not defined.

Authors: We accept the suggestion and add modifications to both the introduction and the suggested line. In addition, a general revision of the text is made in order to try to clarify other confusing statements.On the other hand, the term "liberal attitudes" is changed to "affirmative attitudes" to avoid confusion.

  • Reviewer: Replace the concept element(s) by variable(s)

Authors: Replaced on line 82.

  • Reviewer: The authors write that the 29 included studies have been published between 1986 and 2019. But they also refer to a study of Green published in 1966. This seems in contradiction with the statement that the relevant studies were published between 1986 and 2019. Furthermore, did the authors scan the grey literature for relevant studies not mentioned in the current search systems?

Authors: The study by Green (1966) is not part of the review. It was mentioned because its results were compared with those obtained by Franzini & Casinelli (1986). To avoid confusion of interpretation it is removed from the manuscript. Regarding the question about searching outside the databases mentioned in the methodological section, we conducted secondary searches in reference lists, websites and organisations and contacted relevant authors to obtain unpublished studies. The process is described in figure 1.

  • Reviewer: a. Line 29 -30): “These people may identify as male and female or reject both identities altogether (Safer & 30 Tangpricha, 2019).” Why are the authors using a negative labeling for persons who identify as other than male or female? Why do they not use a positive formulation.

Authors: The suggestion is accepted, as follows: “These people may identify as male and female or identify outside the gender binary.”

Reviewer: (6) b. In close connection with 6a: why do the authors not refer to the SOC-8 (WPATH, 2022), which focus explicitly on language (e.g. transgender and gender divers people) and on the attitudes of health professionals.

Authors: We accept the suggestion and apologize for the exclusion, it is added to the bibliography and the terms are modified according to this document. 

  • Reviewer: Line 41 – 45: “In addition to the barriers to access to services and the perceived abuse by the collective, a series of problems are added, such as high rates of psychological suffering, as well as depression and anxiety of the collective (Bacigalupe et al., 2020; Grant et al., 2011; Latham, 2019; Safer & Tangpricha,2019a; Suess Schwend, 2020; Tateno et al., 2019).” The information in the quoted lines is correct, but, given the aims of the study, not relevant (in the opinion of this reviewer). So it is given in consideration to delete these lines or the clarify how this content is relevant for the aims of the reported review.

Authors: Following the reviewer's suggestion, this extract is deleted.

  • Reviewer: Line 85-88: ““trans*” OR “transgendered person” OR “transgender person” OR “gender dysphoria” OR “gender non-conforming” AND “attitudes” OR “beliefs” OR “practices” OR “stereotyped” OR “knowledge” AND “psychologist” OR “psychotherapist” OR “psychiatrist” OR “nursing” OR “mental health services” OR “mental health professionals” Why didn’t the authors include terms as for example non-binary – queer -gender diverse – gender incongruence. Please clarify and discuss the implications of the made choices in the discussion.

Authors: The search strategy mentioned in the text between lines 85-88 is a summary/extract of the full search strategy, too extensive to be included in the body of the article, so it has been included in a separate table, and sent as an annex, as recommended by PRISMA. In the manuscript it should appear after the above paragraph and it reports the use of other terms in addition to those described, including those proposed by the reviewer.

  • Reviewer: “No temporal or language restrictions were established, given the limitation of similar studies”.

Authors: We precisely indicated that all studies that met the inclusion criteria were included, with no restrictions by language or year of publication. Studies published in different languages were therefore included in the search and translated if necessary, so as not to exclude any studies that might be relevant.

  • Reviewer: Who were the two independent reviewers? Where they members of the research/writing team? Who was the third reviewer? Was this person an outsider or a member of the research team?

Authors: The research team consists of five researchers. In order to comply with the requirements of the PRISMA statement, two of these researchers conducted the search independently of each other, following the predefined strategy, in order to avoid bias in the search and selection process. Two lists of selected studies were drawn up and subsequently agreed upon until a single list was achieved. A similar process was followed for the selection process and bias analysis. Efforts were made to combine the components of each of these peer review teams so that they did not overlap. The last author of the article was considered in all cases as a third reviewer in each of these phases.

  • A.
  • 12) Reviewer: Line 97: “mental health professionals (medicine, nursing, psychology, social

work, and occupational therapy”. Why did you include medicine and nursing as mental health professionals? An endocrinologist for example prescribing gender affirmative hormones is certainly a medical specialist, but not a mental health professional. What are the implications of your definition of mental health professional on the studies that were included or excluded in your review?

Authors: In line 96-97 of the manuscript it is specified as inclusion criteria: (a) the study group includes mental health professionals (medicine, nursing, psychology, social work, and occupational therapy) who are active in related facilities. In other words, the study focuses on analysing the attitudes of mental health professionals, including those working in mental health units or services. Endocrinologists, although they provide care to transgender patients during the transition process, mainly in hormone treatment, are not part of mental health units or facilities.

  • Reviewer: Line 98: “who are active in related facilities”. It is not clear what you mean by related facilities; please clarify.

Authors: We focus on mental health professionals working in this area. We replace it in the text with "specialized centers".

  • Reviewer: Line 146-147: In the case of the qualitative studies, the categories obtained in the different studies were grouped into common themes from which the narrative synthesis of the results was developed.” How were the common themes determined?

Authors: To determine the common themes after coding, we proceeded by means of an inductive-deductive process (Fereday and Muir-Cochrane, 2006). We started from the segmentation and coding of the text after reading and analysis. The codes were then grouped by theme, these themes arising from both the deductive analysis and the objective proposed in the study. In the data analysis section we complete the information provided in the analysis process.

  • Reviewer: Line 173: “The duration of the interventions varied between 3 months and 4 years.” This is a very cryptic sentence; please clarify what you mean.

Authors: We are grateful for the reviewer's suggestion. The wording is clarified to read as follows: "The duration of the interventions evaluated by the studies included in the systematic review ranged from 3 months to 4 years."

  • Reviewer: Line 176: “Given the high heterogeneity of the outcome measures used in the included studies, we decided to group them into categories for analysis.” By whom and how was this grouping done? The current text suggests that it was done arbitralily or in a random way by the authors. Please clarify.

Authors: Thank you for your suggestion, which will be clarified in the text. The sentence should read as follows: "Given the high heterogeneity of the outcome measures used in the included studies, we decided to group them into categories for analysis. The following categories were established that correspond to the main and secondary objectives set out in our study".

The categories on the basis of which the narrative synthesis is made were agreed by the research team, and defined to respond to the main and secondary objectives proposed by our study (lines 71-75), which in turn are related to the outcome measures used in the individual studies included (table 1). The aim is to group the information provided by these outcome measures, which are compared between the different studies and respond to the initially stated objectives.

  • Reviewer: Line 201-203: “Five cross-sectional studies explored the influence of sociodemographic characteristics on professionals' attitudes towards trans people (Dispenza & O’Hara, 2016; Gaspodini & Falcke, 2018a; Johnson & Federman, 2014; Riggs & Bartholomaeus, 2016a; Vijay et al., 2018).” Please add information on HOW these studies explored the influence of sociodemographic variables. In general more attention to methodological aspects of the included studies would be welcome, given the important implications for the (limitations) of generalizability of the results. For example to what extent are the used samples in the included studies WEIRD?

Authors: The methodological designs of each study are indicated in the corresponding tables of results, so as not to excessively dimension the text of the manuscript. Specifically, with respect to the quantitative studies, the methodology is reflected in the first column (table 1). In addition, as indicated in the methodological section, a prior peer review bias analysis was performed to exclude from the review those studies that included serious methodological biases.

  • Reviewer: Line 207: “having been born after 1969”. This reviewer does not understand (how) this conclusion (was reached). Please clarify.

Authors: This conclusion is reviewed. Among the studies evaluated, 5 assessed sociodemographic variables. Of these, only 1 study (Johnson & Federman, 2014) found more favourable attitudes among psychologists born after 1969. Another 3 studies assessed this variable but found no significant results with respect to age. For this reason, this sentence is replaced in the text and reads as follows:  “Regarding age, most studies found no significant results (9,45,46). Only one study(39) found that psychologists born after 1969 had affirmative attitudes and more specific training”.

  • Reviewer: Line 213: “residing in a politically conservative region”. What do you mean by a politically conservative region (and how was this measured). To give an example: does a region refer to countries, cities or …

Authors: This conclusion is drawn from the Johnson & Federman (2014)  study. This study was conducted in the USA, which categorised professionals according to the majority ideology of the state in which they lived.  Another study, Gaspodini et al. (2018) compared attitudes between professionals living in cities and those living in rural areas (interior of Brazil) but found no significant differences.  In accordance with the editor's suggestions, clarifications are added to the text.

  • Reviewer: Line 219 : “noted that the professionals positively valued the training provided and were more open to changing their behaviors after receiving it”. To what kind of training are you referring? This reviewer suggests that you specify the nature of training in Table 1.

Authors: Information on training is added in table 1.

  • Reviewer: Line 233: “Willoughby et al. (2010) found acceptable knowledge among the professionals evaluated” What is acceptable knowledge? Please clarify.

Authors: Clarification added in the text.

“Willoughby et al. (2010) found acceptable knowledge among the professionals evaluated (i.e. the test on knowledge of transgenderism yielded a positively skewed distribution,with scores falling in the higher range, indicating higher knowledge of transgenderism).”

  • Reviewer: Line 272-273 : “The methods of analysis used were thematic analysis (N =), grounded theory (N =), Weiss approximation method (N = ), content analysis (N =), and phenomenological analysis (N =). Please add the number.

Authors: We appreciate the suggestion, and amend the text to read as follows: “The methods of analysis used were thematic analysis (N = 2), grounded theory (N = 4), Weiss approximation method (N = 1), content analysis (N =3), and phenomenological analysis (N = 1).

  • Line 276-277 : “Given the high heterogeneity of the categories established by the studies included in the review, we decided to group them into conceptual themes, as recommended by Percy et al. (2015).” How and by whom was this grouping done?

Authors: Similarly to what was mentioned in item 16 of this review, the categories on which the narrative synthesis is based were agreed upon by the research team, and defined to address the main and secondary objectives proposed by our study (lines 71-75), which in turn relate to the outcome measures employed in the individual qualitative studies included (table 2).

  • Line 303-305: “Four studies directly or indirectly explored the training and knowledge of the professionals (Clark et al., 2017; Gaspodini & Falcke, 2018a; Salpietro et al., 2019; Torres et al., 2015) stressing the importance of training in diversity and multiculturalism to reduce structural barriers in the health services”. Describe also shortly HOW this exploration was done. This remarks is relevant at many places in the manuscript (cfr. the description of the results for the different topics).

Authors: We are grateful for the reviewer's suggestion and proceed to describe in each case the methodology used to achieve these results. The specific information can be found in table 2.

  • Line 329- 333: “Torres et al (2015) added to this the difficulties for children or adolescents to know about the health system and the need to improve the clinical environments by acquiring services such as inclusive bathrooms and information about the resources available to trans people. Both Torres et al. (2015) and Tishelman et al. (2019) highlighted as a structural barrier the cost of access to hormonal treatment and fertility preservation in the U.S.” In Tables 1 and 2 it is NOT specified (if the reviewer read the Tables correctly) what the developmental age was of the groups to which attitudes of mental health professionals were researched. Please add information on this important point. Were the attitudes of mental health professionals studied in relation to children, adolescents, adults, old transgender persons…

Authors: We appreciate the suggestion, which may help to improve the understanding of the results and implications. However, most of the articles reviewed do not include such a specification. The corresponding information has been added to the table in those articles that refer to children, adolescents or youth.

  • Reviewer: On line 308-309 refer the authors still to the SOC-7; please use the SOC-8.

Authors: We are grateful for the reviewer's suggestion and have modified the citation. We revised the text to modify erroneous terms.

  • Reviewer: The discussion is rather superficial and limited in scope. What for example about cultural biases in the studies done so far? How do the authors interpret the meaning of the reported gender differences. Is there a theoretical interpretation of the results? Or is the only (possible) conclusion: this are descriptively the founded results…

Authors: We are grateful for the indication and proceed to revise the discussion to give it greater depth in the sense indicated.

  • Reviewer: Do the authors have NO recommendations to improve the quantitative research approach?

Authors: Indeed, the manuscript only makes suggestions for qualitative research, neglecting the importance of developing quantitative studies. This part is expanded in the discussion by adding the following section:

“Most of the quantitative studies reviewed reflect investigations in the presence of negative or positive attitudes and demographic variables that influence it. Instead, very few provided the components that build a positive or negative attitude. While several of the authors included in this review have attempted to investigate associations between demographic variables and attitudes, the impact on the interaction between TGD and mental health professionals is neglected and requires further investigation. In addition, it would be interesting to use homogeneous instruments and outcome measures that allow meta-analysis of the results.

On the other hand, we recommend…”

  • Reviewer: The authors write on Line 409-412:” On another hand, we propose that health institutions should encourage certain measures in the different care facilities: the provision of public toilets and other environmental actions; the incorporation of experts into the different teams, who can supervise or advise other professionals; the incorporation of people belonging to these minority groups.” Although this is a sympathetic recommendation, it does NOT belong in the discussion given the fact that it is not a consequence/implication of the reported results!

Authors: We are grateful for the suggestion. Indeed, as you indicate, this recommendation, although it follows from the conclusions provided by some of the authors reviewed, does not respond to the objectives initially set out, so we have proceeded to remove it from the text.

  • Reviewer: In Line 418 the authors use “measures”. Do they mean measures (in the sense of measurement instruments) or interventions? If they refer to measurement instruments, this is new information which does not belong in the discussion (given the current manuscript).

Authors: The term "measures" is used in the indicated text as a synonym for "actions" or "interventions". We replace it in the manuscript as follows:

“Thus, more studies are needed to try to understand what interventions are indispensable to reduce the differences perceived by trans people when demanding psychiatric or psychological treatment in the Public Health System and the impact these actions have on the modification of health perceived by trans people”. 

  • Reviewer: Please specify in the abstract the number of studies for qualitative, quantitative and mixed studies.

Authors: The required information is included in the abstract.

  • Reviewer: The phrase “Recent years” in the abstract is not informative. Please be more precise.

Author: The expression is changed to "The last decades", more in line with the terms used in the introduction.

Thank you very much for your review of our article, which has enabled us to improve the manuscript considerably. Best regards.

Reviewer 2 Report

This systematic review topic is timely and important. However, the manuscript needs revision to address some important topics described below.

1. METHODS: Why did the systematic analysis not include studies published more recently (2020-2022)? This is a manuscript submitted in 2023, and more recent publications are pivotal to better explaining mental health professionals’ attitudes toward trans people. On May 25, 2019, the 11th revision of ICD-11 was launched, officially removing diagnoses associated with transgender persons from the chapter on mental and behavioural disorders. This is a key change and might somehow influence mental health professionals’ practices, behaviours and perceptions of transgender patients. This reviewer highly recommends updating the review.

2. METHODS: Another concern is related to the large timeframe selected by the authors: 1986 to 2019. Many cultural and social changes happened during those 33 years, the training of mental health professionals also changed a lot, and some acceptable practices adopted during the 80s and 90s by mental health professionals are nowadays considered human rights violations of LGBTQ+ patients. Could the authors elaborate on this concern?

3. RESULTS: Although the idea of including quantitative, qualitative and mixed-methods studies seems highly interesting, the results are presented in a shallow and inadequate way. It is important to describe more in deep the SR findings. Perhaps even separating the studies within specific timeframes, to account for the huge changes experienced in the mental health field during the last three decades. After reading the results, many questions remained unanswered: is there a specific pattern by region? Where the studies were conducted? Did you include studies from low and middle-income countries? If you did, is there any difference when you compare professionals from different regions? Or is there any specific characteristic more common within a certain profession? The practices, training and overall relationship with patients are quite singular for selected mental health professionals working in medicine vs. nursing vs. psychology vs. social work vs. occupational therapy… I understand that a carefully designed and conducted systematic review takes time and a lot of effort. But the key goal is to synthesize in deep all available research that is relevant to your specific research question. The results, as presented in this manuscript, falls short of delivering what we expected. After reading the manuscript, I am left without a proper answer to the proposed question: “What attitudes do mental health service professionals have towards trans people?”… A thorough review of the results section is needed.

4. RESULTS: Figure 1, Tables 1 and 2 are missing from the submission.

5. DISCUSSION: When the authors summarize their findings, the following sentence was included: 

“The individual factors that have been related to the professionals' positive attitudes were: being a woman, having previous experiences with trans people, having received specific training during the university stage, or belonging to any type of minority” 

This sentence doesn’t answer the proposed research question. It is doubtful that those broad characteristics are predictors of having a ‘positive’ attitude toward trans patients: being a woman, belonging to any (!) type of minority, and having received training while in university…

Transgender and gender nonconforming persons have a higher prevalence of mental disorders and interact with mental health care systems and mental health professionals at high rates when compared to the general population. But yet, until nowadays, transgender and gender nonconforming persons experience substantial barriers to accessing the mental health care they need. This systematic review could help highlight where are the struggles, from healthcare professionals’ perspectives. But the results and discussion sections lack the necessary depth to properly address and try to respond to the proposed research question.

6. Some sentences are just too vague to understand. For instance, in lines, 230-231, “Francia-Martínez et al. (2017) concluded that a high number of professionals are unaware of important aspects of the problems of this group” what important aspects? What problems of ‘this group’? Are you referring to mental health disorders among transgender and gender non-conforming persons? If you are, please rework your sentence to make it clear.

Another example of an unclear sentence (lines 296-297): “Carrizo (2014) highlighted that the professionals interviewed related the term “gender” to sexist violence and reproductive health.”  What do you mean by ‘sexist violence’? And why is gender written with quotation marks?

There are other sentences like those, please read your manuscript again, trying to make it clear. The manuscript also needs a second round of spelling and grammar review. There are some sentences in Spanish, please correct them.

Thanks for the opportunity to review your work. I look forward to seeing a revised version of it.

Author Response

Regarding the manuscript: Attitudes and beliefs of Mental Health Professionals towards Trans People: a systematic review of the literature.

Manuscript ID:  ijerph-2237561

Dear reviewer, we deeply appreciate the review of the manuscript, which we believe will help to significantly improve the quality of our manuscript.

We will then proceed to respond as best we can to your suggestions, indications and requests for clarification. We have arranged our responses in the same order that they were listed in your review:

  1. REVIEWER: METHODS: Why did the systematic analysis not include studies published more recently (2020-2022)? This is a manuscript submitted in 2023, and more recent publications are pivotal to better explaining mental health professionals’ attitudes toward trans people. On May 25, 2019, the 11th revision of ICD-11 was launched, officially removing diagnoses associated with transgender persons from the chapter on mental and behavioural disorders. This is a key change and might somehow influence mental health professionals’ practices, behaviours and perceptions of transgender patients. This reviewer highly recommends updating the review.

AUTHORS: The systematic review actually includes studies published up to June 2021, when the databases were last accessed (line 91). However, no studies from 2021 were included as those published up to that time did not meet the specified inclusion criteria or did not pass the risk of bias analysis.

We appreciate, however, the reviewer's indication, as we believe that a further update of the search up to the current date could incorporate studies of interest and enrich the review. Therefore, the submitted manuscript has been updated to more recent dates and the search has been extended to March 2023.

  1. REVIEWER: METHODS: Another concern is related to the large timeframe selected by the authors: 1986 to 2019. Many cultural and social changes happened during those 33 years, the training of mental health professionals also changed a lot, and some acceptable practices adopted during the 80s and 90s by mental health professionals are nowadays considered human rights violations of LGBTQ+ patients. Could the authors elaborate on this concern?

AUTHORS: No time limits were set on the search in order to collect the maximum number of relevant studies. However, the largest number of relevant studies related to our objective were published after 2010 (n=28), while only 1 study (Franzini & Casinelli, 1986) was found from earlier dates. For this reason, cultural and social changes prior to 2010 could not be adequately assessed. At this point, the research team debated the appropriateness of including the 1986 study. However, by consensus of the authors, it was decided to include it due to its importance.

  1. RESULTS: Although the idea of including quantitative, qualitative and mixed-methods studies seems highly interesting, the results are presented in a shallow and inadequate way. It is important to describe more in deep the SR findings. Perhaps even separating the studies within specific timeframes, to account for the huge changes experienced in the mental health field during the last three decades. After reading the results, many questions remained unanswered: is there a specific pattern by region? Where the studies were conducted? Did you include studies from low and middle-income countries? If you did, is there any difference when you compare professionals from different regions? Or is there any specific characteristic more common within a certain profession? The practices, training and overall relationship with patients are quite singular for selected mental health professionals working in medicine vs. nursing vs. psychology vs. social work vs. occupational therapy… I understand that a carefully designed and conducted systematic review takes time and a lot of effort. But the key goal is to synthesize in deep all available research that is relevant to your specific research question. The results, as presented in this manuscript, falls short of delivering what we expected. After reading the manuscript, I am left without a proper answer to the proposed question: “What attitudes do mental health service professionals have towards trans people?”… A thorough review of the results section is needed.

AUTHORS: Some of the questions posed by the reviewer appear in tables 1 and 2, which he apparently did not receive correctly as indicated in point 4. They indicate the countries in which each study was conducted, year of publication, characteristics of the sample, etc.

With regard to the question posed by the reviewer and which constitutes the main objective of our study: "What attitudes do mental health service professionals have towards trans people?" We recognise that in the results section we only show the results obtained. However, lines 186 (quantitative studies) and 297 (qualitative studies) address this question, which is more clearly stated in the conclusions section (lines 423-431). The difficulty encountered by the reviewer could be due to the variability of the measurement instruments used in the studies (Hill & Willoughby, 2005; Kattari et al., 2018; Walch et al., 2012; Willoughby et al., 2010) that tend to assess the difference between positive and negative attitudes without further elaboration, which does not allow us to go deeper. The new version of the manuscript incorporates this issue in the section: implications for the practice and research.

On the other hand, we are grateful for the recommendation made by the reviewer and the results section has been revised in accordance with his indications (we have added a clarification in the results on differences by geographical area. We also changed the order and put the section on attitudes and beliefs about trans people first).

  1. RESULTS: Figure 1, Tables 1 and 2 are missing from the submission.

AUTHORS: We regret the error in the submission. Please find attached tables and figure 1 in the revised manuscript.

  1. DISCUSSION: When the authors summarize their findings, the following sentence was included: 

“The individual factors that have been related to the professionals' positive attitudes were: being a woman, having previous experiences with trans people, having received specific training during the university stage, or belonging to any type of minority” 

This sentence doesn’t answer the proposed research question. It is doubtful that those broad characteristics are predictors of having a ‘positive’ attitude toward trans patients: being a woman, belonging to any (!) type of minority, and having received training while in university…

Transgender and gender nonconforming persons have a higher prevalence of mental disorders and interact with mental health care systems and mental health professionals at high rates when compared to the general population. But yet, until nowadays, transgender and gender nonconforming persons experience substantial barriers to accessing the mental health care they need. This systematic review could help highlight where are the struggles, from healthcare professionals’ perspectives. But the results and discussion sections lack the necessary depth to properly address and try to respond to the proposed research question.

 AUTHORS: With regard to the first question posed by the reviewer, this statement certainly does not answer the PICO question: "What attitudes do mental health service professionals have towards trans people?

However, as a secondary objective we pose (lines 73-75): which elements are related to the attitudes of these professionals, paying special attention to their psychosocial, training, and cultural aspects?" On the other hand, the statement "The individual factors that have been related to the professionals' positive attitudes were: being a woman, having previous experiences with trans people, having received specific training during the university stage, or belonging to any type of minority" has been revised in the manuscript and the information that could mislead the reader has been removed.

With regard to the content of the discussion, we are grateful for the reviewer's suggestion, which we consider relevant, and we will therefore proceed to modify the discussion taking into account the aspects he mentions.

  1. REVIEWER: Some sentences are just too vague to understand. For instance, in lines, 230-231, “Francia-Martínez et al. (2017) concluded that a high number of professionals are unaware of important aspects of the problems of this group” what important aspects? What problems of ‘this group’? Are you referring to mental health disorders among transgender and gender non-conforming persons? If you are, please rework your sentence to make it clear.

Another example of an unclear sentence (lines 296-297): “Carrizo (2014) highlighted that the professionals interviewed related the term “gender” to sexist violence and reproductive health.”  What do you mean by ‘sexist violence’? And why is gender written with quotation marks?

There are other sentences like those, please read your manuscript again, trying to make it clear. The manuscript also needs a second round of spelling and grammar review. There are some sentences in Spanish, please correct them.

AUTHORS: We are grateful for the reviewer's indications and a clarification of the sentences indicated, as well as a complete revision of the text in the sense indicated.

Thank you for revising our article, which has allowed us to improve the manuscript considerably. Best regards.

Reviewer 3 Report

Thank you for inviting me to review the manuscript. I congratulate the authors, they have undertaken an important topic, I have no comments

Author Response

Dear reviewer, thank you for the work done and the comments included in the process. We appreciate the very positive considerations made about the text and especially the time you have spent reading and analyzing the manuscript.

Round 2

Reviewer 1 Report

First of all, thanks to the authors for taken the comments of this reviewer seriously and for processing them adequately. Just a few minor problems remain before this reviewer will propose acceptance to the editor:

(1) Line 12: "The last decades scientific research on transpeople  has increased considerably". Delete this general, superficial and non informative sentence. Start instead with the core business of this review: "A systematic review was conducted...."

(2) Line 24-25: "The importance of training in gender diversity is confirmed". In the opinion of this reviewer this conclusion is not a result of the review as such, but a recommendation by the authors. So, take into consideration the following formulation: "In line with the results of our review, we recommend that training of professionals is important to improve their positive attitudes towards transgender and gender diverse people". [In general, this reviewer does not think that the methodological quality of the reviewed studies allows making causal inferences; however, it is the impression of this reviewer that the authors have a different view on this. Nevertheless, please check once again if you have made causal inferences were the data do not justify such inferences. If you would stick to your causal interpretations after such a check, I will not object as a reviewer; but will still object as a reader to your "to strong"causal inferences).

(3)Line 34 -51: This is a very broad introduction, which is not much focused upon the core variables of this review. This reviewer suggest to focus much more and solely on the core relevant variables: attitudes of mental health professionals towards transgender and gender diverse persons. In consequence this reviewers advice is to shorten this paragraph considerably.

(4)Line 68: Insert towards transgender and gender diverse people after "attitudes and beliefs of professionals"

(5)Line 86: PICO. Maybe, it is better to write (only the first time when you use the term PICO " PICO (orP(atient/population), Intervention, Comparison, Outcomes).  

(6)Line 110-114: Inclusion- and exclusion criteria. This reviewer still does not understand why for example medicine and nursing are classified as mental health professionals. They ARE NOT. See also the first review on this point. And the answer of the authors did not solve this problem (for this reviewer).

(7)Line 113: "specialized centers". Remains rather vague: do you mean mental health and/or gender affirming treatment centers?

(8)Line 149: "The articles selected for our study". Replace study by review.

(9)Replace Line 241-244 by the sentence: "The sociodemographic characteristics ....were correlated with positive attitudes"

(10)Line 393: Replace study by review

(11)Line 418-419: "as it comes from a limited number of observational studies" Is only the number a problem or is also the quality of the designs a problem? If the quality of the (used) designs is also a problem (as is the view of this reviewer) please make also a recommendation about the quality of the designs to be used.

(12)Line 420: The authors use the term political positioning, but do not define it. In consequence, the meaning of this phrase remains unclear for the reader. Please clarify.

(13)Line 429: "Our review points to different proposals". The recommendations are the recommendations of the authors (based on the results of their review), but are not recommendations of the review as such. Rephrase for example as: : "Based on the results of our review, we propose....)

(14)Line 451: "formulation of affirmative responses". Please replace by: application of affirmative responses

(15)Line 456: "our review proposes". Replace by:  We propose....

(16)Limitations and strenghts: What about cultural (positioning) limitations in the research done so far (and by consequence in your review)?

(17)Line 483-485: this is a very broad, strong and general  conclusion. Is it based on your data? If not, delete it.

(18)Line 515: Insert after professionals: towards transgender and gender diverse people.

Author Response

Regarding the manuscript: Attitudes and beliefs of Mental Health Professionals towards Trans People: a systematic review of the literature.

Manuscript ID:  ijerph-2237561

Thank you for the opportunity to revise the above-mentioned manuscript. We thank the reviewer for the pertinent remarks and suggestions that we believe have helped to improve the manuscript.

Please find a detailed description below of how we have responded to each of them. We have also attached a revised manuscript, where changes are marked up using the “track

changes” function.

1) REVIEWER Line 12: "The last decades scientific research on transpeople has increased considerably". Delete this general, superficial and non informative sentence. Start instead with the core business of this review: "A systematic review was conducted...."

AUTHORS: Thank you for the suggestion. The sentence is deleted as indicated in the revised manuscript.

(2) REVIEWER. Line 24-25: "The importance of training in gender diversity is confirmed". In the opinion of this reviewer this conclusion is not a result of the review as such, but a recommendation by the authors. So, take into consideration the following formulation: "In line with the results of our review, we recommend that training of professionals is important to improve their positive attitudes towards transgender and gender diverse people". [In general, this reviewer does not think that the methodological quality of the reviewed studies allows making causal inferences; however, it is the impression of this reviewer that the authors have a different view on this. Nevertheless, please check once again if you have made causal inferences were the data do not justify such inferences. If you would stick to your causal interpretations after such a check, I will not object as a reviewer; but will still object as a reader to your "to strong"causal inferences).

AUTHORS: The reviewer's suggestion is taken into account. Although the text has been revised to reduce the presumption of causality, some sentences, such as the one noted above, may still require modification in this regard. The revised manuscript incorporates the reviewer's proposal.

(3)REVIEWER. Line 34 -51: This is a very broad introduction, which is not much focused upon the core variables of this review. This reviewer suggest to focus much more and solely on the core relevant variables: attitudes of mental health professionals towards transgender and gender diverse persons. In consequence this reviewers advice is to shorten this paragraph considerably.

AUTHORS: We are grateful for the recommendation, which we incorporate into the revised manuscript, where aspects not related to the attitudes of mental health professionals are removed.

(4)REVIEWER. Line 68: Insert towards transgender and gender diverse people after "attitudes and beliefs of professionals"

AUTHORS: Thank you for your appreciation. The proposed extract from the revised manuscript is inserted.

(5) REVIEWER. Line 86: PICO. Maybe, it is better to write (only the first time when you use the term PICO " PICO (orP(atient/population), Intervention, Comparison, Outcomes).  

AUTHORS: Thank you for the suggestion, which is incorporated into the revised manuscript.

(6)REVIEWER. Line 110-114: Inclusion- and exclusion criteria. This reviewer still does not understand why for example medicine and nursing are classified as mental health professionals. They ARE NOT. See also the first review on this point. And the answer of the authors did not solve this problem (for this reviewer).

AUTHORS: I apologise if this point was not sufficiently clarified in the first revision. With regard to Medicine, we included those medical professionals with the speciality of Psychiatry and who worked in mental health institutions or organisations. In Spain, as in other countries, Psychiatry is a speciality of Medicine. With regard to nursing, something similar occurs, we included nursing professionals with the speciality of mental health or equivalent in their country, and who carried out their functions in mental health centres or agencies, as in the previous case. This is clarified in the revised manuscript.  More information can be found at the following link: https://www.nami.org/About-Mental-Illness/Treatments/Types-of-Mental-Health-Professionals

(7)REVIEWER: Line 113: "specialized centers". Remains rather vague: do you mean mental health and/or gender affirming treatment centers?

AUTHORS: Indeed, it was still somewhat vague, as we were referring to mental health centres, institutions or services. This is likely to have been due to a translation error. It is clarified in the revised manuscript.

(8) REVIEWER. Line 149: "The articles selected for our study". Replace study by review.

AUTHORS: Thank you for the indication. It is amended in the revised manuscript.

(9)Replace Line 241-244 by the sentence: "The sociodemographic characteristics ....were correlated with positive attitudes"

AUTHORS: It is modified in the manuscript.

(10)Line 393: Replace study by review

AUTHORS: The word in the manuscript is modified

(11)Line 418-419: "as it comes from a limited number of observational studies" Is only the number a problem or is also the quality of the designs a problem? If the quality of the (used) designs is also a problem (as is the view of this reviewer) please make also a recommendation about the quality of the designs to be used.

AUTHORS: Thank you for the recommendation, a clarification is added in this regard.

(12) REVIEWER Line 420: The authors use the term political positioning, but do not define it. In consequence, the meaning of this phrase remains unclear for the reader. Please clarify.

AUTHORS: The sentence is clarified (line 379-380 in the revised manuscript).

(13)REVIEWER Line 429: "Our review points to different proposals". The recommendations are the recommendations of the authors (based on the results of their review), but are not recommendations of the review as such. Rephrase for example as: : "Based on the results of our review, we propose....)

AUTHORS: The sentence in the text is modified.

(14) REVIEWER. Line 451: "formulation of affirmative responses". Please replace by: application of affirmative responses

AUTHORS: The reviewer's suggestion is accepted and the sentence is modified in the text.

(15)REVIEWER. Line 456: "our review proposes". Replace by:  We propose....

AUTHORS: Amended in the text according to the reviewer's suggestion.

(16) REVIEWER. Limitations and strenghts: What about cultural (positioning) limitations in the research done so far (and by consequence in your review)?

AUTHORS: The following paragraph is added to the text:

“There are several limitations to the data collated that warrant further discussion. First, the quantitative and qualitative data collected in this review relate predominantly to a female, white, heterosexual, Western population, with studies conducted mostly in the USA. Although data are collected from research conducted in non-Western regions, it is not possible to make a comparison between results due to the variability of measurement tools employed. This reduces the generalisability of the results obtained and urges caution in interpreting the results. The inability to ascertain the views of under-represented groups (non-white populations and sexual/ethnic minorities) may reflect a difficulty in recruiting these populations or methodological flaws in the design of the studies reviewed.”

(17)REVIEWER Line 483-485: this is a very broad, strong and general  conclusion. Is it based on your data? If not, delete it.

AUTHORS: This conclusion is deleted as suggested by the reviewer.

(18) REVIEWER Line 515: Insert after professionals: towards transgender and gender diverse people.

AUTHORS: The sentence in the text is modified.

Thank you for revising our article, which has allowed us to improve the manuscript considerably. Best regards.
